# Influence of skill relatedness on the location choice of heterogeneous labor force in Chinese prefecture-level cities

**Xiaoqi Zhou**[1,2], **Rongjun Ao**[1,2], **Yierfanjiang Aihemaitijiang**[1,2]*, **Jing Chen**[1,2], **Hui Tang**[1,2,3]

**1** Key Laboratory for Geographical Process Analysis & Simulation Hubei province, Central China Normal University, Wuhan, Hubei, China, **2** College of Urban and Environmental Science, Central China Normal University, Wuhan, Hubei, China, **3** School of Architecture and Urban Planning, Hunan City University, Yiyang, Hunan, China

* erfan818@163.com

**Data Availability Statement:** All data files are available from the figshare database (https://doi.org/10.6084/m9.figshare.22883072.v1).

## Abstract

High-quality regional development should be promoted by facilitating inter-regional mobility of heterogeneous labor force to optimize its spatial allocation. This study incorporates skill relatedness into spatial categorization and selection effects, and explores how skill-relatedness affects the location choice of heterogeneous labor force. To do so, we use labor force migration data and employee data by occupation subcategory from the 2000 National Population Census and 2015 National Population Sample Survey. The empirical evidence provides three major findings. First, there are significant regional differences in labor migration rates by the occupational group between cities in China, and the trend is increasing. Regional concentration of location choice is increasing and six significant agglomerations are formed. Second, skill relatedness positively affects the location choice of the heterogeneous labor force in Chinese cities. When cities' skill-relatedness is more robust, influence on labor location choice is more remarkable. In cities with high-size classes, the effect of high-skill relatedness on labor location choice is higher. Third, labor force with solid skill relatedness with regional employment moves to the location owing to the spatial sorting effect. Labor force without skill relatedness or weak relatedness moves out or does not move to the location owing to the spatial selection effect.

## Introduction

At present, China's economy has shifted from the high-speed growth stage to the high-quality development stage [1]. The root of high-quality development lies in the economy's vitality, innovation, and competitiveness. On the one hand, high-quality development should ensure steady development. On the other hand, such a development must focus on the efficiency of economic growth (i.e., resource allocation efficiency). Therefore, improving the efficiency of resource allocation is an important step in promoting high-quality development [2]. Human capital is a crucial determinant in fostering innovation [3]. Moreover, the importance of its rational resource allocation has become increasingly prominent in the context of the

**Funding:** This research was funded by the National Natural Science Foundation of China (Grant numbers 42271188). Our funders played a role in study design, data collection and manuscript preparation.

**Competing interests:** The authors have declared that no competing interests exist.

continuous decrease in labor supply, uneven spatial distribution of the labor force, and considerably significant regional differences in skill structure in China [4–6]. Spatial allocation of labor resources is realized through the locational choice of the labor force. Since the 2000s, the overall education level of China's labor force has been increasing [7], migratory population has increased and expanded [8]; and population mobility has gradually changed from spontaneous mobility under homogeneity to autonomous mobility under heterogeneity [9]. Therefore, smooth inter-regional mobility of a heterogeneous labor force is important for optimizing regional employment skill structure, promoting optimal spatial allocation of the labor force, and promoting high-quality regional development. Recently, many Chinese cities have frequently introduced preferential new policies for talent, resulting in competition for talent resources. However, not all cities benefit from these new talent policies [10]. Accordingly, such results have prompted scholars to consider further the laws of heterogeneous labor location selection.

Relatedness is often defined as the similarity between firms, products, or industries in terms of technology, management, factors of production, and infrastructure [11]. Relatedness is considered by evolutionary economic geography as an important law for studying economic evolutionary processes [12]. In recent years, scholars have extended relatedness to the analysis of regional employment skill structure by combining the research progress of labor mobility and occupational skills to establish the concept of skill relatedness [13]. In general, research on skill-relatedness remains in its initial stage. Only a relatively few papers have mainly focused on the measurement of skill relatedness [14], impact of skill relatedness on regional occupational progression [15, 16], and employment scale change [17, 18]. Muneepeerakul et al. [16] first assessed the impact of occupational linkages on occupational progression in US metropolitan areas. Neffke et al. [13] established method to measure inter-industry skill relatedness using inter-industry labor mobility data from Sweden to calculate inter-industry skill relatedness and explain the impact of skill relatedness on industrial evolution. Alabdulkareem et al. [19] used data on occupations and their skill structures provided by the US Department of Labor O*NET to directly calculate the correlations between skills and establish a skilled space. Chinese scholars have only recently started to focus on skill relatedness. Nevertheless, skill-relatedness is more often briefly introduced and discussed from the perspective of inter-industry labor mobility and human capital [20, 21], and empirical studies still need to be revised. He et al. [20] introduced the research progress of skill relatedness from European scholars. Xu et al. [22] constructed China's first urban skill space network, revealing the positive correlation between skill relatedness and intercity migration of college-graduated labor.

Geographic studies of labor location choice are often macroscopic, taking the regional labor force as a whole and emphasizing the influence of exogenous geographic and environmental factors. On the bases of classical migration theories, such as push–pull theory, differential income theory, cost–benefit theory, expected income theory, and new economic theory of migration [23–25], studies have been conducted to investigate the effects of economic and non-economic factors, such as environmental pollution [26, 27], household status [28], income level [29, 30], employment opportunities [31, 32], and amenity variables [30] on labor location choice. Meanwhile, the perspective of labor economics is markedly microscopic and often takes the individual labor force as object of study. Accordingly, labor economics scholars have observed the heterogeneity of migrating labor earlier [33]. Spatial economics has expanded the study of location choice of the heterogeneous labor force and gradually formed an analytical framework with spatial class division [34, 35] and selection [36, 37] as the core. This area has become the theoretical cornerstone of the study of the heterogeneous labor force's location choice. Venables [38] found that the self-selection mechanism of the labor force leads to a high-skilled labor force in cities with high cost of living. Eeckhout et al. [36] noted that owing

to skill complementarities, high- and low-skilled labor is classed as large cities and medium-skilled labor as small cities. In general, class allocation and selection is a spontaneous process of spatial allocation owing to the pairing of labor and cities. The correlation between the skill attributes of the labor force and skill structure of pre-existing employment in the city and the matching environment determine the locational decisions of the heterogeneous labor force [36]. Note that despite the differences between geography and spatial economics in interpreting the location choice of the heterogeneous labor force, scholars commonly agree that distance is the core factor of location choice.

Although scholars have made significant progress in the research on skill-relatedness and labor location choice, there are still some limitations. First, the research on skill relatedness is in its initial stage, and the research content is relatively single [14, 20, 21]. The few studies on labor mobility have only taken labor mobility between industries as the basic index to measure the skill relatedness between industries [21]. Hence, there is a need for additional research on the influence of skill relatedness on the interregional mobility of heterogeneous labor. Second, most of the studies have only explored the influencing factors of interregional mobility of heterogeneous labor force by simply dividing the labor force into two (high and low) and three (high and low) skill level groups, lacking a considerably microscopic consideration of labor force heterogeneity [36, 38]. Geography examines the influencing factors of labor migration more from the perspective of exogenous differences between regions [23–25]. Although spatial economics proposes spatial class classification and selection as the theoretical cornerstone of heterogeneous labor location decisions, the related research has mainly focused on their regional economic effects. Empirical analysis is needed on how spatial class classification and selection endogenously affect heterogeneous labor location decisions [34–37]. Therefore, there is a need to study skill relatedness and heterogeneous labor location choices in the case of China. Given that China is the largest developing country and an emerging economy in transition, it has provided the best laboratory for empirical research since the turn of the century. Finally, most of the existing studies in China have been conducted at the provincial level [27]. Moreover, the location of the heterogeneous labor force is mainly city scale, thereby necessitating an empirical analysis at the city level.

For analysis, this paper used labor migration data and employed person data by sub-category of occupation by city and region provided by the 2000 National Population Census and 2015 National Population Sample Survey micro-data sets in China. We used the analysis of the spatial characteristics of heterogeneous labor location choices in China as basis to explore whether or not there is a correlation between skill relatedness and urban labor location choices. If there is a correlation, then the following questions must be answered: How does skill relatedness endogenously and differentially affect the location choice of a skilled heterogeneous labor force? The results of our study can enrich the research on skill relatedness and location choice of the heterogeneous labor force, provide a reference for decision-making to optimize the spatial allocation of labor resources, and improve the efficiency of labor market matching in different regions. Our study also has the following contributions. 1) Skill heterogeneity should consider the difference between high and low skill levels and variability of abilities required by occupations. The types of labor force classified in this paper are more diversified, which is a more microscopic consideration of labor force heterogeneity. 2) With the help of skill relatedness, this study attempts to substantially explain how spatial class classification and selection endogenously affect the location choice of a heterogeneous labor force. 3) The empirical analysis uses Chinese prefecture-level cities as spatial units to avoid the problem of disregarding the spatial heterogeneity of small regional categories caused by substantially large research scale. 4) Lastly, this study considers whether or not the effect of skill relatedness on heterogeneous labor migration differs among cities of different size classes.

## Research hypotheses

In the classical migration model, the distance between the place of migration and departure is the resistance to labor migration. Distance is a multidimensional concept with extensive connotations [39]. In addition to geographical distance, it involves socio-cultural, institutional, informational, and cognitive distance [40–42]. The principle of relatedness emphasizes the influence of relatedness or proximity on the evolution of economic space [21]. Consistent that firm entry or exit is influenced by multidimensional proximity, the locational choice of labor is also influenced by multidimensional proximity [43]. Migrants tend to choose incoming locations that are considerably close and have similar environments to save costs and avoid risks [44, 45]. Skill proximity is an important component of multidimensional proximity. Given the uncertainty of the matching process and effects of skill distance between labor and urban job markets, skill proximity has also become an important factor in the location choice of heterogeneous labor [46]. Skill proximity is not a concept of physical spatial proximity in the same way as geographical distance; it reflects the correlation between the skill attributes of the individual labor force and skill structure of regional employment (i.e., skill relatedness). The stronger the skill relatedness, the higher the skill proximity and the closer the skill distance. In addition, owing to the uneven economic development in China, the industrial structure and employment structure of cities in different size classes vary significantly [47]. Moreover, the correlation between the skill attributes of the individual labor force and employment structure of cities in different size classes may be different. As such, Hypotheses 1 and 2 are proposed.

Hypothesis 1: Skill relatedness positively affects the location choice of the heterogeneous labor force in Chinese prefecture-level cities. The effect on the location choice of the labor force is greater when the skill-relatedness of cities is stronger.

Hypothesis 2: The magnitude of the effect of skill relatedness on labor location choice differs among cities of different size classes.

The analytical framework centred on spatial sorting, and selection effect can be used to analyze the location choice of a heterogeneous labor force. The drivers of spatial sorting and selection of heterogeneous subjects include various economic and non-economic factors [26–32]. The push-pull theory framework also consists of the regional push and pull factors. Only the push-pull factors under the push-pull theory are consistent for all labor forces. In contrast, spatial class sorting and choice theory can consider the variability of push-pull factors for different labor force individuals. The impact of skill relatedness on skill heterogeneous labor location choice can also be interpreted with the help of this analytical framework. In particular, the spatial sorting effect refers to the choice of cities by micro-subjects, an ex-ante locational choice made by heterogeneous subjects based on their talents [36, 37]. The spatial selection effect is the city's choice of micro issues, manifested by eliminating subjects with lower ability owing to market competition or cost of living, leaving the most talented ones behind afterwards [34, 38]. The effect of skill relatedness on the locational choice of the heterogeneous labor force in Chinese prefecture-level cities is shown in Fig 1. Suppose the labor force is related (similar or complementary) to a city's existing employment skill structure. In the case of increased relevant employment opportunities in that city, the labor force may migrate to that city based on the matching skills they possess, thus meeting the city's labor demand. In cases such as industrial structure changes, when the skill attributes of the city's existing workforce do not match the regional labor market demand, there will be a workforce that is priced out of the market and chooses to leave the city because it cannot find a job. The effect of spatial sorting and selection is a spontaneous process of spatial allocation caused by the pairing between the labor force and cities. Accordingly, Hypothesis 3 is proposed.

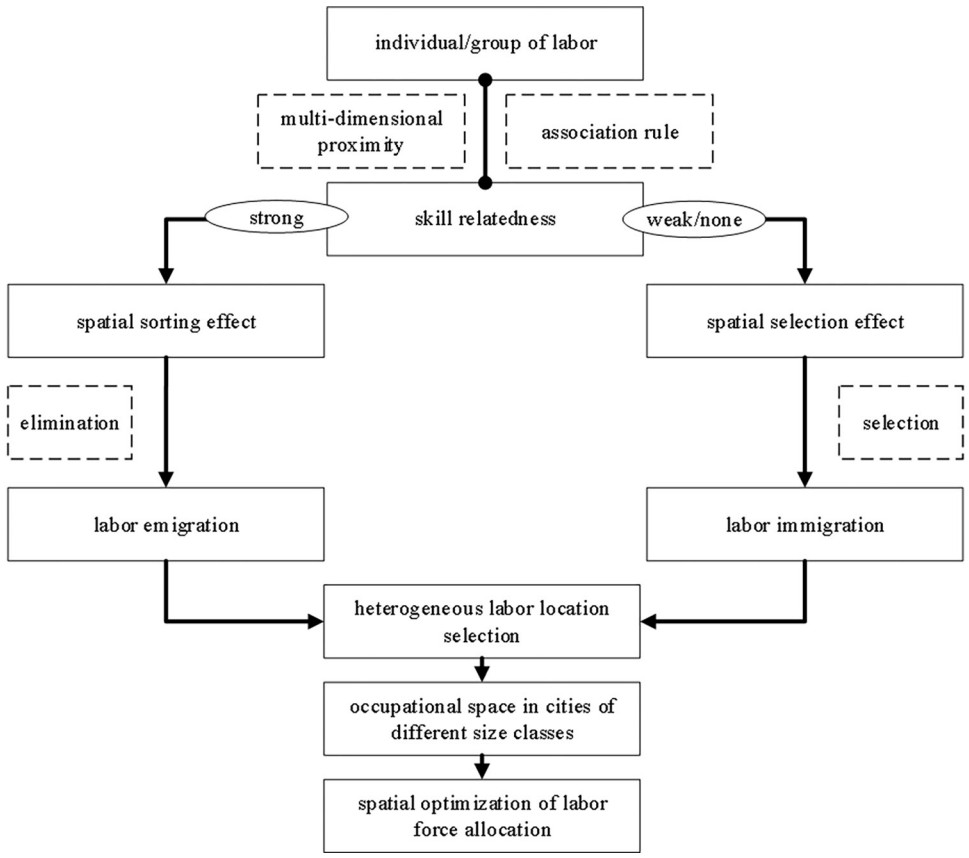

**Fig 1. Effect of skill relatedness on the locational choice of the heterogeneous labor force.**

Hypothesis 3: The labor force with skill relatedness or strong relatedness with regional employment moves into the location owing to the spatial sorting effect. The labor force without skill relatedness or weak relatedness moves out or does not move into the location owing to the spatial selection effect.

## Materials and methods

### Study area

The spatial unit was prefecture-level cities, considering that cities are the main concentration of occupations with different skill levels and considering data availability. To ensure the dynamic comparability of spatial data, this study uniformly divided prefecture-level cities according to the 2015 caliber. The current research juxtaposed four municipalities directly under the central government with prefecture-level cities as spatial units. Lastly, the research scope of this paper included 288 prefecture-level cities in China (Hong Kong, Macao, and Taiwan are excluded) (Fig 2).

### Data description and data source

Skills are not uniformly defined. Although educational attainment has been used to measure skill level, a growing number of researchers believe that educational attainment is not the best indicator of skills and propose to measure skills by occupation. Ingram et al. [48] point out that years of education are a rough measure of skills because a degree is not equivalent to the

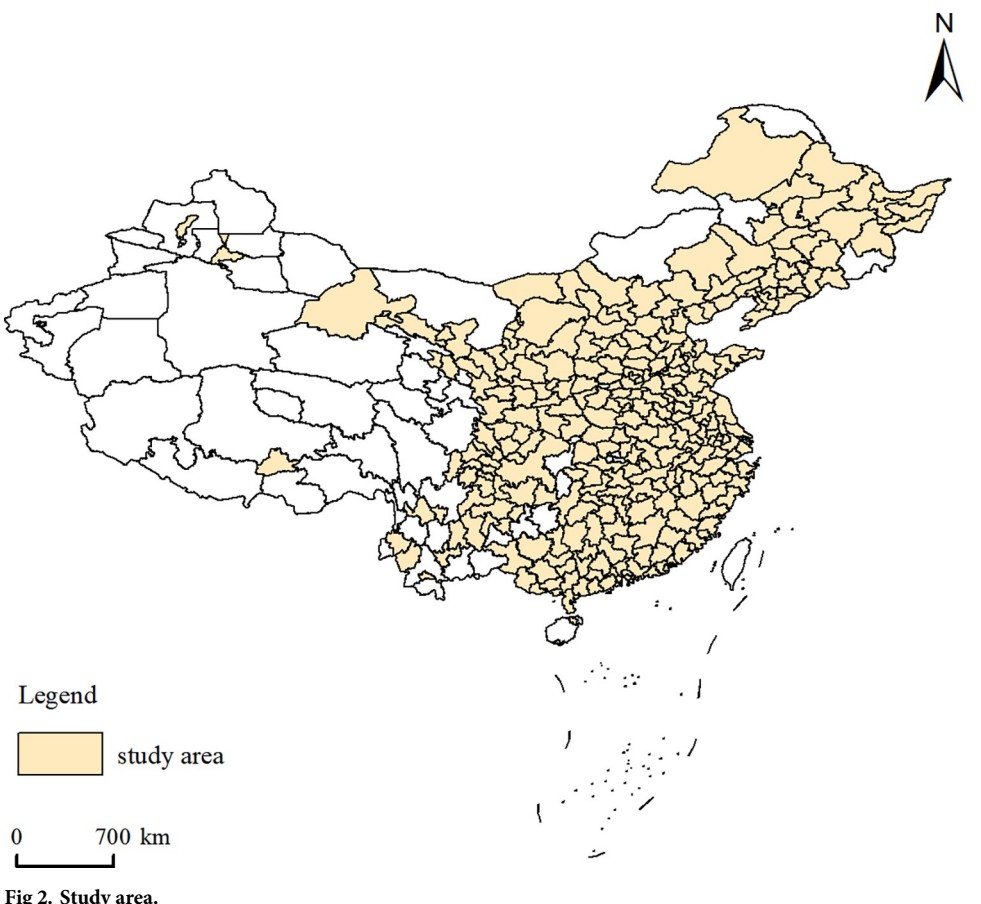

**Fig 2. Study area.**

talent it implies, and all students, even those who graduate from the same school with the same degree, may not graduate at the same level. Florida et al. [49] argue that formal education can only measure potential talents or skills, while occupations provide an idea of "how human talents or abilities are absorbed and utilized by the economy." Based on existing research, this paper uses occupation as the primary indicator to measure skills, considering that skills reflect the ability to complete a job.

Given the availability of data, the primary core data in this paper (i.e., number of employed persons in prefecture-level cities by occupational subcategory and the number of employed persons who moved in) were obtained from the 2000 China Population Census and 2015 China Population Sample Survey microdata sets.

In the International Standard Classification of Occupations (ISCO-08) issued by the International Labor Organization in 2008, occupations and skills are divided in correspondence [50]. An occupation was defined as a group of jobs with a high degree of similarity in major tasks and duties. A skill is an ability to perform the tasks and duties of a particular job. The People's Republic of China Classification of Occupations (2015 Edition) draws on the advanced experience of international occupational classifications, and the classification caliber is considerably similar to ISCO-08 [51]. Classification standards used in the Chinese census and sample survey are generally consistent with ISCO-08. Considering the purpose of this paper and availability of data, the occupational subcategory is uniformly clustered into 13 occupational categories. Following Acemoglu et al. [52] and Consoli et al. [53], we clustered the 13 occupational groups into three skill levels based on each occupational group's overall

**Table 1. Clustering approach for the occupation categories.**

| Skill levels | High-skilled | Medium-skilled | Low-skilled |
|---|---|---|---|
| Occupation categories | Managers | Office personnel | Delivery service workers |
| | Professionals | Commercial sales workers | Residential service workers |
| | Technicians | Mechanical and electrical assemblers | Agricultural production workers |
| | | Production and operation workers | |
| | | Mining and construction workers | |
| | | Transportation operators | |
| | | Safety and security workers | |

job nature and job skill requirements. Agricultural and service-oriented jobs were considered low-skilled occupations, routine jobs as medium-skilled occupations, and abstract jobs as high-skilled occupations (Table 1).

The China 2000 Population Census data and China 2015 Population Sample Survey micro-data set provided data on the migration of employees by occupation subcategories for each prefecture-level city. The labor force migration data by skill type were obtained by aggregating according to the clustering scheme in Table 1. On the bases of the indicators of "time of moving to the current place of residence" and "previous city of residence" in the 2000 data set, migration data within prefecture-level cities were excluded, and migration data across prefecture-level cities were obtained. On the basis of the "time of coming to live in the city" indicator in the 2015 data set, we obtained the in-migration data of the prefecture-level cities in 2015. Considering that the data set is the result of surveys conducted at five-year intervals and that the single-year data values for individual regions are considerably small, this study included the labor migration data accumulated in the past five years as labor migration.

The climatic suitability values involved in this study were measured using the humidity–heat Index (HI), drawing on the calculation method of Feng et al. [54], and HI was obtained from the National Earth System Science Data Center (http://www.geodata.cn/). Other socio-economic and cultural data were obtained from the *China City Statistical Yearbook*, *China Statistical Yearbook for Regional Economy*, and statistical yearbooks of the province or prefecture-level city.

## Research methodology

**Skill relatedness measure.** This paper explores the relationship between skill relatedness and the location choice of the heterogeneous labor force in Chinese cities, requiring obtaining skill relatedness between occupational category and city. This study draws on the calculation methods of Hidalgo et al. [55] and Boschma et al. [56].

First, using the locational quotient method, a matrix of city-occupational classes (n × k) consisting of 288 cities and 13 occupational classes was constructed to identify occupational classes with specialization advantages. The level of specialization of occupational class $i$ in city $c$ is as follows:

$$RCA_{i,C,t} = \frac{lab_{i,c,t}/\sum_i lab_{i,c,t}}{\sum_c lab_{i,c,t}/\sum_{c,i} lab_{i,c,t}} \tag{1}$$

where $RCA_{i,c,t}$ denotes the specialization level of city $c$ in occupational class $i$ in period $t$ and $lab_{i,c,t}$ denotes the number of employees in occupational class $i$ in city $c$ in period $t$. If $RCA \geq 1$, it means that the number of employees in occupation class $i$ of city $c$ is higher than the average number of employees in occupation class $i$ of all cities and has the advantage of specialization.

Second, skill relatedness between occupational groups was calculated. The calculation formula is as follows:

$$\varphi_{i,j,t} = min\{P(x_{i,t}|x_{j,t}), P(x_{j,t}|x_{i,t})\}, x_{i,c,t} = \begin{cases} 1, RCA_{i,c,t} \geq 1 \\ 0, RCA_{i,c,t} < 1 \end{cases} \quad (2)$$

where $\varphi_{i,j,t}$ denotes skill relatedness between occupation class $i$ and $j$ in period $t$. The value ranges between 0 and 1; the larger the value, the stronger the skill correlation between the two occupation classes. $P(x_{i,t}|x_{j,t})$ indicates the probability that occupation class $j$ has the advantage of specialization. Occupation class $i$ also has the advantage of specialization in period $t$. The specific calculation is expressed by the ratio of the number of cities where occupation class $i$ and $j$ have the advantage of specialization and the number of cities where occupation class $i$ has the advantage of specialization. When $RCA_{i,c,t} \geq 1$, $x_{i,c,t}$ takes the value of 1. When $RCA_{i,c,t} < 1$, $x_{i,c,t}$ takes the value of 0.

Third, skill relatedness between occupation class $i$ and city $c$ was calculated using the following formula:

$$SRD_{i,c,t} = (\sum_{i \neq j} \varphi_{i,j,t} \cdot x_{i,c,t}) / \sum_{i \neq j} \varphi_{i,j,t} \quad (3)$$

where $SRD_{i,c,t}$ denotes skill relatedness between occupational class $i$ and city $c$ in period $t$, $\varphi_{i,j,t}$ denotes skill relatedness between occupational class $i$ and occupational class $j$ in period $t$, $\sum_{i \neq j} \varphi_{i,j,t} \cdot x_{i,c,t}$ is the correlation matrix between occupational class $i$ and occupational class $j$ in period $t$ and relatedness between occupational class $i$ and occupational class $j$ in city $c$ in period $t$, respectively ($RCA_{i,c,t} \geq 1$, and 0 when $RCA_{i,c,t} < 1$) and summed up by the product of the logical value of whether or not class $i$ is a specialized occupational class. The larger value of $SRD_{i,c,t}$ indicates that the occupational class $i$ of city $c$ is associated with more specialized occupational classes in period $t$ and has a higher probability of becoming a specialized occupational class of city $c$ in the future. In this study, for the criteria for classifying correlations in existing studies [55–57] and combining available data, skill relatedness ($\varphi_{i,j,t}$) $\geq 0.55$ were considered strong correlations, $0.4 \leq \varphi_{i,j,t} < 0.55$ as correlations exist but are not significant, and $\varphi_{i,j,t} < 0.4$ as no correlations.

## Heterogeneous labor migration rate measurement

The locational choice of a heterogeneous labor force is mainly characterized by the migration rate of the labor force [58].

Labor migration rate between occupation class $i$ and city $c$ is as follows:

$$Migrate_{i,c,t} = migration_{i,c,t} / local_{i,c,t} \quad (4)$$

where $Migration_{i,c,t}$ denotes labor migration of occupation class $i$ in city $c$ in period $t$ and $local_{i,c,t}$ denotes the number of local workers of occupation class $i$ in city $c$ in period $t$. When analyzing the spatial distribution of heterogeneous labor migration rate, this paper explores the regional concentration of labor migration rate with the help of coefficient of variation and Gini coefficient; the specific formula can be referred to the related literature [59].

## Empirical test of the effect of skill relatedness on heterogeneous labor location choice

First, using prefecture-level cities as the observed sample, correlation analysis is performed to examine the correlation between average city skill relatedness and urban labor migration rate

in 2000 and 2015 in terms of the full sample, different skill relatedness degrees, and different city sizes. The full sample includes annual and multi-year correlations. This paper conducts instantaneous correlation analysis for cross-sectional data in 2000 and 2015 and multi-year correlation analysis for panel data formed by pooling data in 2000 and 2015. Second, to reveal the relationship between different skill relatedness degrees and urban labor migration rates, correlation analysis is conducted after dividing skill relatedness degrees into three intervals: strong skill relatedness, skill relatedness present but insignificant, and no correlation. Lastly, correlation between high skill relatedness and urban labor migration rate is tested for differences in cities of different size classes. This paper conducts correlation analysis using SPSS software and calculates the following formula:

$$r = \frac{\sum_{i=1}^{n}(x_i - \bar{x})(y_i - \bar{y})}{(n-1)S_x S_y} \tag{5}$$

where $r$ is the correlation coefficient and $n$ is the number of prefecture-level cities, $x_i$ is the skill correlation of city $i$, $y_i$ is the labor migration rate of city $i$, $\bar{x}$ is the mean value of skill correlation of all cities, $\bar{y}$ is the mean value of labor migration rate of all cities, and $S_x$ and $S_y$ are the standard deviations of the $x$ and $y$ variables, respectively, of all cities.

A multiple regression model is used to test the effect of skill relatedness on the location choice of a heterogeneous labor force, using occupational category as the observed sample and prefecture-level city as the spatial unit. Labor migration rate is the dependent variable and skill relatedness degree is the core independent variable. In addition, many factors affect heterogeneous labor location choice, and we control for several important factors. According to the existing hypotheses, the spatial sorting effect is an *ex-ante* heterogeneous locational choice made by heterogeneous subjects based on the complementarity of their talents and city size [36, 37]. Cities with more suitable natural environments, more developed economies, and higher incomes are more likely to be chosen by skilled labor [27, 29, 60]. Therefore, we introduce four variables, namely, climate suitability, green coverage, GDP per capita, and average wage, as control variables in the test of the class separation mechanism. Climate suitability is measured using HI. Green coverage of built-up areas measures green coverage. GDP per capita is measured by the ratio of GDP to the resident population in urban areas. Lastly, average wage of urban workers on the job measures the average wage. The spatial selection effect is the city's selection of micro subjects, with the core idea that fierce market competition and higher cost of living will eliminate the less capable labor force [34, 38]. Therefore, we introduce three variables, namely, employment competition, consumption expenditure, and cost of living, as control variables for the selection mechanism test. Employment competition is measured by the labor supply and demand ratio in urban areas. Consumption expenditure is measured by urban residents' per capita consumption expenditure, and the average sales price of commodity houses measures the cost of living. This paper conducts regression analysis using Stata17 software. For model selection, the ordinary least squares method (OLS) is used to evaluate the relationship between two or more factor attributes and minimize the sum of squares of residuals of all observed values in the selected global regression model. In this study, variables were first standardized to eliminate the multicollinearity problem, and then the OLS model was used to analyze the cross-section data of 2000 and 2015 preliminarily. Then, a two-factor fixed effects model was used for the panel data consisting of 288 cities in both years. This study used the F, and Hausman tests to determine specific model selection when establishing the panel data model. Since the P-value of the F test was 0, the Hausman test accepted the null hypothesis that explanatory variables were related to individual (or time) effects, so the fixed effects model was chosen. Moreover, the two-factor fixed effects model was selected because of the highest

confidence and goodness of fit of the estimated results of the two-factor fixed effects model. The regression equation is as follows.

$$Crosssection_{c,t} = \alpha_0 + \alpha_1 SRD_{c,t} + \alpha_2\ln(qh_{c,t}) + \alpha_3\ln(lh_{c,t}) + \alpha_4\ln(jy_{c,t}) + \alpha_5\ln(rj_{c,t}) + \alpha_6\ln(gz_{c,t})$$
$$+ \alpha_7\ln(xf_{c,t}) + \alpha_8\ln(jz_{c,t}) + \delta_c + \varepsilon_{c,t} \tag{6}$$

$$PanelData_{i,c,t} = \beta_0 + \beta_1 SRD_{i,c,t} + \beta_2\ln(qh_{c,t}) + \beta_3\ln(lh_{c,t}) + \beta_4\ln(jy_{c,t}) + \beta_5\ln(rj_{c,t})$$
$$+ \beta_6\ln(gz_{c,t}) + \beta_7\ln(xf_{c,t}) + \beta_8\ln(jz_{c,t}) + \mu_t + \delta_i + \varepsilon_{i,c,t} \tag{7}$$

where *Crosssection* and *PanelData* represent the regression equations for cross-sectional and panel data, respectively, *i* represents occupational class, *c* represents prefecture-level city, and *t* represents the year. In addition, *SRD* stands for skill relevance, measured by the skill relevance of the occupational category to the city; *qh* stands for climate suitability; *lh* stands for green coverage; *jy* stands for employment competition; *rj* stands for GDP per capita; *gz* stands for average wage; *xf* stands for consumption expenditure; *jz* stands for housing cost; *μ* stands for time fixed effect; *δ* stands for individual fixed effect; and *ε* represents the random disturbance term.

## Results

### Spatial and temporal evolution of heterogeneous labor location choices in prefecture-level cities in China

**Spatial and temporal evolution of labor migration rates in prefecture-level cities.** The map visual representation of labor force migration rates of Chinese prefecture-level cities formed a map of the spatial and temporal evolution of labor force migration rates of Chinese prefecture-level cities from 2000 to 2015 (Fig 3). Fig 3 shows that the mobility of the labor force in China significantly increased from 2000 to 2015, and the regional differences in labor force migration rates between prefecture-level cities are significant and increasing. The variation and Gini coefficients of labor migration rates between prefecture-level cities in China

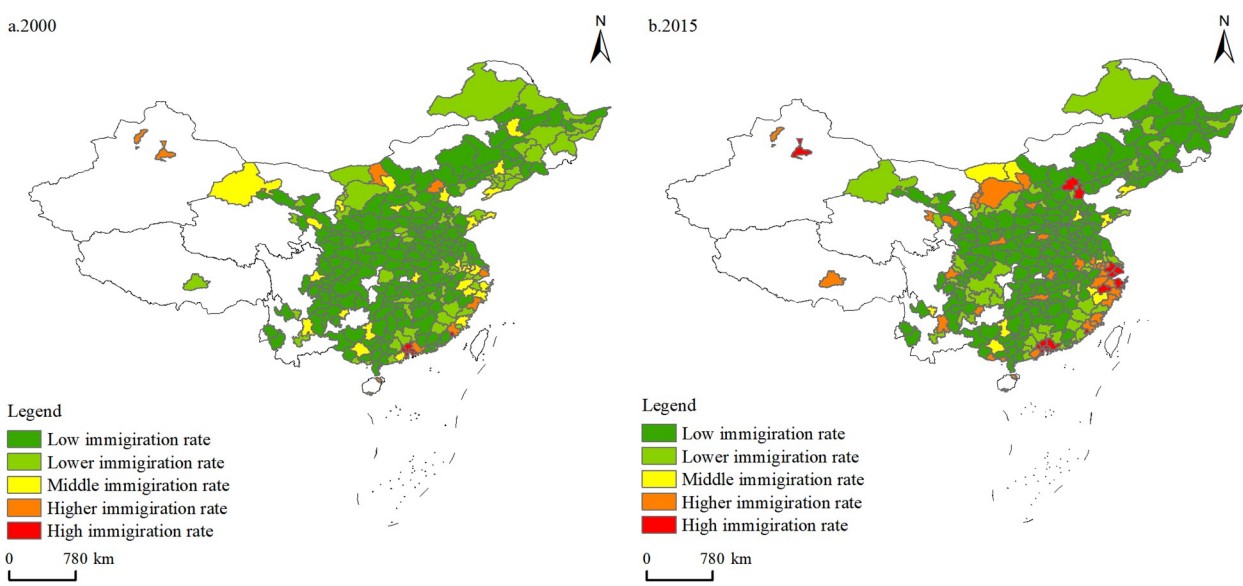

**Fig 3. Distribution of immigration rate of labor force between prefecture-level cities in China.**

**Table 2. Coefficients of variation and Gini coefficients of labor migration rates by occupational category in 2000 and 2015.**

| | 2000 | | 2015 | |
|---|---|---|---|---|
| | Coefficients of variation | Gini coefficients | Coefficients of variation | Gini coefficients |
| Managers | 0.4103 | 0.2214 | 1.8732 | 0.6943 |
| Professionals | 0.3571 | 0.1875 | 1.3162 | 0.5925 |
| Technicians | 0.6349 | 0.3010 | 1.3626 | 0.6301 |
| Office personnel | 0.3470 | 0.1777 | 1.4169 | 0.6338 |
| Commercial sales workers | 0.4480 | 0.2273 | 1.1355 | 0.5222 |
| Mechanical and electrical assemblers | 0.6427 | 0.2816 | 1.6045 | 0.6681 |
| Production and operation workers | 0.6331 | 0.2946 | 1.4488 | 0.6204 |
| Mining and construction workers | 0.7430 | 0.3702 | 1.6264 | 0.6706 |
| Transportation operators | 0.5816 | 0.2685 | 2.3073 | 0.8226 |
| Safety and security workers | 0.4591 | 0.2333 | 1.8272 | 0.7550 |
| Delivery service workers | 0.5682 | 0.2713 | 1.5875 | 0.6581 |
| Residential service workers | 0.4343 | 0.2222 | 1.3127 | 0.5942 |
| Agricultural production workers | 1.3335 | 0.3724 | 3.6443 | 0.8132 |

expanded from 0.9714 and 0.4010 in 2000 to 1.5833 and 0.6152 in 2015, respectively. Note that expanding regional differences implies an increase in the regional concentration of labor location choices. In particular, the labor force has continuously concentrated in the more developed socioeconomic cities in the Bohai Sea Rim, Yangtze River Delta, West Coast of Taiwan, and Pearl River Delta regions on the eastern coast; and the provincial capitals and some industrial and mining cities in the central and western regions.

**Spatial and temporal evolution of labor migration by occupational category in prefecture-level cities.** Coefficients of variation and Gini coefficients of labor migration rates for each occupational category between local cities were calculated. Table 2 shows the results. The average labor force migration rates of each occupational category in 31 provinces in China in 2000 and 2015 were calculated. Fig 4 shows the results.

From 2000 to 2015, regional differences in labor migration rates of each occupational category between prefecture-level cities show a widening trend. Coefficient of variation and Gini coefficient of the labor migration rate of each occupational category between prefecture-level cities increased. The degree of regional concentration of labor force location selection in each occupational category increased, and six major concentration areas were formed: Beijing-based Bohai Sea Cluster, Shanghai-based Yangtze River Delta Cluster, West Coast Cluster, Pearl River Delta Cluster, Hainan Cluster, and Xinjiang and Tibet-based Western Cluster. The number of cities with high and higher migration increased in some occupational categories, such as managers, technicians, commercial sales workers, mechanical and electrical assemblers, production and operation workers, mining and construction workers, transportation operators, delivery service workers, and agricultural production workers. This result indicates that the high migration of these occupational labor forces into the agglomeration area presents an expanding trend. Moreover, there are some occupational categories of high migration, and high migration to the city number decreased, such as professionals, office personnel, safety and security workers, and residential service workers. This result indicates that the occupation of labor force migration to a few cities was concentrated.

In 2000, Beijing was the main location of choice for each occupational group in the Bohai Rim agglomeration, with high and higher-migration rates for professionals, commercial sales workers, production and operation workers, mining and construction workers, safety and security workers, residential service workers. Shanghai was the core of the Yangtze River Delta

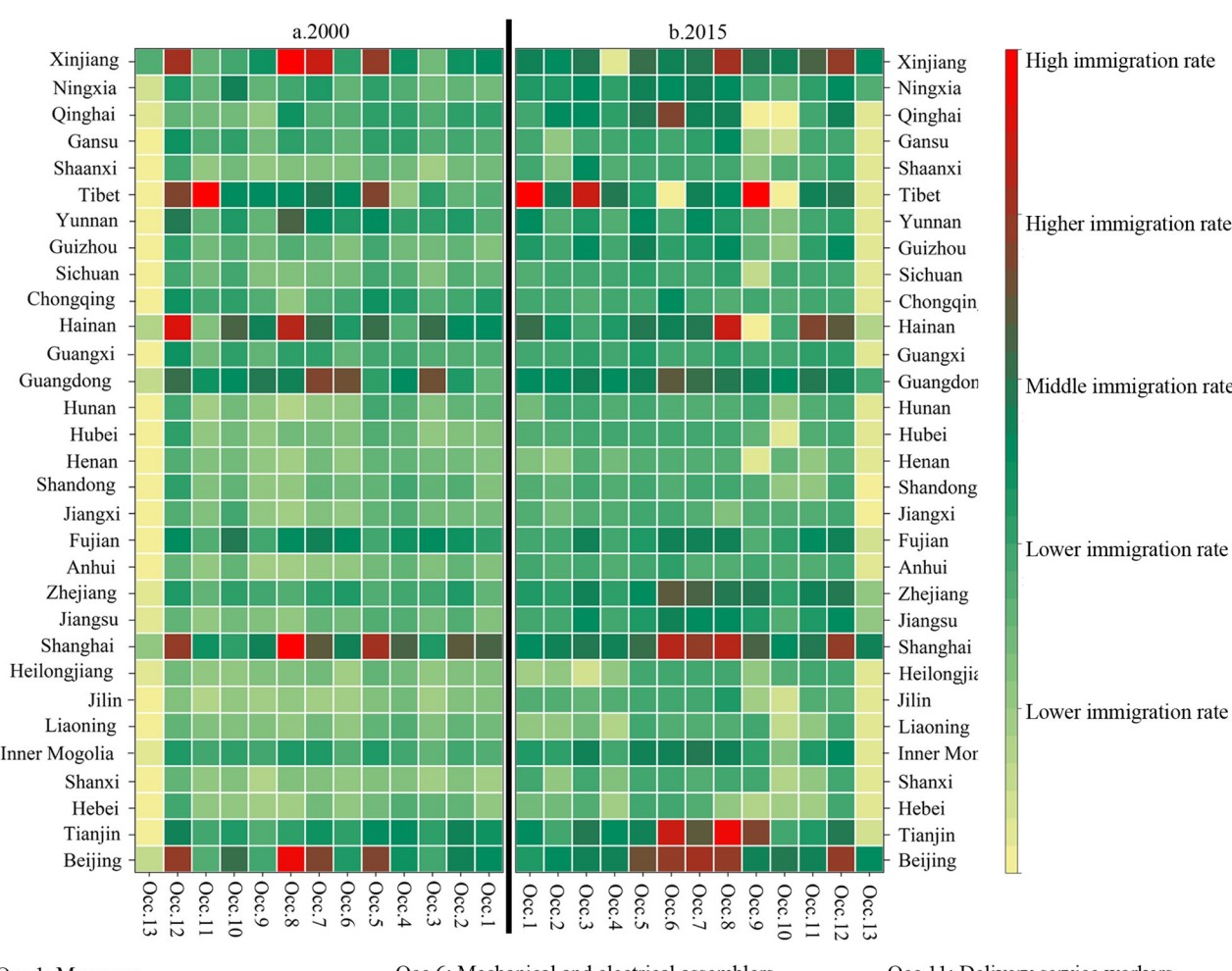

**Fig 4. Average migration rate of labor by occupation category in 31 provinces in China in 2000 and 2015.**

agglomeration, with high and higher-migration rates for managers, professionals, office personnel, commercial sales workers, electrical and mechanical assemblers, production and operation workers, mining and construction workers, transportation operators, and residential service workers. The agglomeration area on the west coast of Taiwan was mainly concentrated in Fujian Province, with Quanzhou, Fuzhou, Xiamen, and Putian as main cities. Technicians, mechanical and electrical assemblers, production and operation workers, and safety and security workers had high and higher-migration rates. In the Pearl River Delta, mainly in Shenzhen, Dongguan, Zhongshan, Guangzhou, and other cities, technicians, mechanical and electrical assemblers, production and operation workers, transportation operators, residential service workers, and other occupational groups have high and higher-migration rates of migration. Hainan agglomeration was mainly in Haikou, Sanya. Technicians, commercial sales workers, production and operation workers, mining and construction workers, and other occupation categories have a high or high immigration rate. The western agglomeration was

dominated by Lhasa and Urumqi, with high and higher-migration rates of commercial sales workers, production and operation workers, mining and construction workers, transportation operators, and residential service workers.

In 2015, regional differences in labor migration rates across occupational groups between prefecture-level cities further expanded, and the regional concentration of location choices increased. In 2015, more sub-occupational labor force categories converged into the six major agglomeration areas. In the Bohai Rim agglomeration, Beijing's occupational groups with high in-migration rates increased. The core region expanded to Tianjin. Technicians, office personnel, mechanical and electrical assemblers, transportation operators, and delivery service workers became new high-in-migration occupational groups. In the Yangtze River Delta agglomeration, migration rate of each occupational group in Shanghai further increased and expanded to Jiangsu and Zhejiang, with technicians and delivery service workers becoming the new high-migration occupational groups. In Hainan Province, migration rate of technicians and safety and security personnel decreased, and migration rate of managers increased. In the western agglomeration, in-migration rate of high-skilled labor, such as managers, professionals, and technicians, increased rapidly. It became the new high in-migration occupational category, while in-migration rate of occupational categories, such as commercial sales workers and mechanical and electrical assemblers, decreased. In Urumqi, migration rate increased for all occupational groups except office personnel.

## Effect of skill relatedness on the location choice of heterogeneous labor force

**Correlation of skill relatedness with urban labor migration rates.** The results of the correlation analysis between skill relatedness and urban labor migration rate were obtained in this study using the SPSS calculations, as shown in Table 3. In the full-sample model, correlation coefficients of the urban average skill relatedness with the urban labor migration rate in the immediate years of 2000 and 2015 are 0.842 and 0.598, respectively. Correlation coefficient of the multi-year pooled 2000–2015 is 0.561. All correlation coefficients are significant at the 1% level. Thus, skill relatedness has a significant positive correlation with the labor force migration rate. That is, skill relatedness positively contributes to the increase of the labor force migration rate. To reveal the relationship between different degrees of skill relatedness and urban labor migration rate, this study divided skill relatedness into three intervals for correlation analysis: strong skill relatedness, skill relatedness existed but not significant, and no relatedness existed. The results show that the labor migration rate increases with the increase of skill relatedness.

**Table 3. Correlation coefficient between skill relatedness and labor migration.**

| | Full-sample | | | Degree of skill relatedness | | | City size class | | |
|---|---|---|---|---|---|---|---|---|---|
| | **2000** | **2015** | **2000∪2015** | **strong skill relatedness** | **skill relatedness existed but not significant** | **no relatedness existed** | **large-scale** | **medium-sized** | **small-sized** |
| **correlation coefficient** | 0.842*** | 0.598*** | 0.561*** | 0.456*** | 0.085 | 0.263*** | 0.444*** | 0.527*** | 0.417** |
| **sample size** | 288 | 288 | 576 | 189 | 75 | 312 | 101 | 56 | 32 |

Note: Large-scale cities refer to cities with a resident population of 1 million and above in urban areas, medium-sized cities with a resident population of 500,000–1 million, and small-sized cities with a resident population of 500,000 and below in urban areas

*** indicates $P < 0.01$

** indicates $P < 0.05$, and

* indicates $P < 0.1$. 2000 ∪ 2015 is the set of urban labor migration rate and urban skill relatedness in 2000 and 2015.

Correlation coefficient is 0.456. When skill relatedness is strong, there is a moderate positive correlation between skill relatedness and labor migration rate. Correlation coefficient is 0.456. When skill relatedness is present but not significant, correlation is also not significant. However, the positive value of the correlation coefficient above zero can still reflect the positive relationship between the two. That is, skill relatedness has a positive promotion effect on labor migration, and this promotion effect increases with the increase of skill relatedness. In addition, correlation coefficients of the non-skill-related samples are above zero. Correlation coefficient of the unskilled sample is below 0.3, showing a weak negative correlation. This result indicates that the unskilled sample does not have a facilitating effect on labor migration and may even hinder labor migration.

To reveal whether or not the relationship between high-skill relatedness and urban labor migration differs among cities of different size classes, this study further divided the sample into three categories, namely, large-scale, medium-sized, and small-sized cities, for correlation analysis. The results show that correlation between high-skill relatedness and labor migration rate varies among cities of different size classes. Correlation coefficient was 0.444 in cities with high-size classes, 0.527 in medium-sized cities, and 0.417 in low-size classes. This result indicates that correlation between high-skill relatedness and labor migration rate was most pronounced in medium-sized cities, followed by cities with high-size class, and weakest in cities with low-size class.

Overall, the results show that skill-relatedness has a significant positive correlation with the labor migration rate of cities. That is, correlation increases with the increase of skill-relatedness. Positive correlation with the labor migration rate is more significant when the skill relatedness of cities is at an extremely high state. This significance is most pronounced in medium-sized cities, followed by large and small cities.

**Impact of skill relatedness on labor migration by occupational category.** The regression results of skill relatedness on labor migration by occupational category were calculated using Stata17, as shown in Table 4. Estimated values of the regression coefficients of skill relatedness

**Table 4. Impact of skill relatedness on labor migration by occupation category.**

|  | full-sample | | | | | |
|---|---|---|---|---|---|---|
|  | **2000** | **2015** | **2000∪2015** | **2000** | **2015** | **2000∪2015** |
| skill relatedness | 0.1037*** (0.0048) | 0.01333*** (0.0065) | 0.0181*** (0.0055) | 0.0513*** (0.0048) | 0.0012 (0.0068) | 0.0206*** (0.0058) |
| climate suitability |  |  |  | -0.0052** (0.0020) | 0.0023 (0.0016) | 0.0032 (0.0023) |
| green cover |  |  |  | -0.0161*** (0.0024) | -0.0224*** (0.0037) | 0.0011 (0.0043) |
| employment competition |  |  |  | -0.0799*** (0.0101) | 0.0416*** (0.0085) | -0.0197** (0.0092) |
| GDP per capita |  |  |  | 0.0161*** (0.0028) | 0.0262*** (0.0031) | -0.0037 (0.0034) |
| average wages |  |  |  | 0.0739*** (0.0086) | 0.0043 (0.0095) | 0.0400*** (0.0077) |
| consumer spending |  |  |  | 0.0454*** (0.0085) | 0.0249*** (0.0090) | 0.0172*** (0.0041) |
| cost of living |  |  |  | 0.0242*** (0.0045) | 0.0753*** (0.0072) | -0.0031 (0.0061) |
| constant | 0.1637*** (0.0040) | -0.0213*** (0.0034) | 0.1811*** (0.0021) | -0.8349*** (0.0824) | -0.2684** (0.1117) | -0.3326*** (0.0758) |
| individual fixed effects | yes | yes | yes | yes | yes | yes |
| time fixed effects | no | no | yes | no | no | yes |
| sample size | 3744 | 3744 | 7488 | 3744 | 3744 | 7488 |
| R-squared | 0.3039 | 0.1876 | 0.7080 | 0.4778 | 0.3531 | 0.7154 |

Note

*** indicates $P < 0.01$

** indicates $P < 0.05$, and

* indicates $P < 0.1$. 2000 ∪ 2015 is the set of urban labor migration rate and urban skill relatedness in 2000 and 2015.

are generally significant at the 1% level. Correlation between skill relatedness and labor migration is relatively robust in the full-sample model. That is, skill relatedness has a significant positive effect on labor migration. Each unit increase in the skill relatedness of an occupational category with a city increases the labor migration rate of that occupational category in that city by 0.1037 and 0.0133 in the same year and increases the labor migration rate by 0.0181 in multiple years. After adding the control variables of the natural environment and socioeconomic dimensions, skill relatedness still has a significant positive effect. Each unit increase in the skill relatedness of an occupational category with a city increases the labor migration rate of that occupational category in that city by 0.0513 and 0.0012 for the immediate year and 0.0206 for the multi-year period. Although the estimated regression coefficient for 2015 is insignificant, its positive value above zero still reflects a positive effect of skill relatedness on labor migration.

To reveal the effects of different skill relatedness on urban labor migration, this study divided skill relatedness into three intervals: strong skill relatedness, skill relatedness present but not significant, and no relatedness for regression analysis (see Table 5). The results show that the promotion effect of skill relatedness on labor migration increases with the increase of skill relatedness. Regression coefficient is insignificant when skill relatedness between the occupational group and the city is strong. However, a positive regression coefficient value above zero can still reflect the positive effect. Regression coefficient of the unskilled relatedness sample shows negative insignificance, but a regression coefficient below zero can also indicate that unskilled relatedness may discourage labor migration. In addition, regression results of the strong skill relatedness based on the labor migration rate of different size cities indicate that the degree of impact of the strong skill relatedness on the labor migration rate varies across cities of different size classes. Regression coefficient estimates of skill relatedness are significant at the 5% level in cities with high-size classes. A one-unit increase in the skill relatedness of an occupational class with a city is associated with a 0.1135% increase in the labor migration rate for that occupational class. Although regression coefficient estimates are

**Table 5. Regression results of different skill relatedness and different urban scales.**

| | degree of skill relatedness | | | city size class | | |
|---|---|---|---|---|---|---|
| | strong skill relatedness | skill relatedness existed | no relatedness existed | large-scale | medium-sized | small-sized |
| skill relatedness | 0.0711** (0.0297) | 0.0320 (0.2126) | -0.0203 (0.0152) | 0.1135** (0.0521) | 0.0149 (0.1207) | 0.0003 (0.1915) |
| climate suitability | 0.0062 (0.0041) | -0.0125 (0.0139) | 0.0043 (0.0031) | 0.0071 (0.0045) | -0.0075 (0.0241) | 0.0203 (0.0680) |
| green cover | -0.0053 (0.0123) | 0.0669*** (0.0211) | -0.0041 (0.0027) | -0.0053 (0.0166) | -0.0221 (0.0406) | 0.0148 (0.0353) |
| employment competition | -0.0664** (0.0258) | -0.0260 (0.0727) | 0.0061 (0.0123) | -0.0914* (0.0469) | -0.0325 (0.0490) | -0.3039 (0.3234) |
| GDP per capita | -0.0194* (0.0028) | 0.0112 (0.0234) | -0.0018 (0.0042) | -0.0260 (0.0161) | 0.0016 (0.0467) | 0.0075 (0.0793) |
| average wages | 0.0812*** (0.0226) | 0.0126 (0.0738) | 0.0184** (0.0076) | 0.0583* (0.0341) | 0.2296*** (0.0698) | 0.1168 (0.2603) |
| consumer spending | 0.0151 (0.0121) | -0.0539 (0.0634) | 0.0112*** (0.0032) | 0.0167 (0.0129) | -0.0624 (0.1716) | 0.0568 (0.2327) |
| cost of living | 0.0445*** (0.0143) | -0.0484 (0.0469) | 0.0064 (0.0064) | 0.0368* (0.0216) | -0.0416 (0.0766) | -0.1122 (0.0909) |
| constant | -0.6095*** (0.2153) | 0.4737 (0.8973) | -0.0885 (0.0751) | -0.4344 (0.3126) | -1.5202 (0.9467) | -1.5396** (0.7329) |
| individual fixed effects | yes | yes | yes | yes | yes | yes |
| time fixed effects | yes | yes | yes | yes | yes | yes |
| sample size | 2500 | 939 | 4049 | 1333 | 730 | 437 |
| R-squared | 0.6911 | 0.7992 | 0.7682 | 0.6833 | 0.7987 | 0.7594 |

Note: Large-scale cities refer to cities with a resident population of 1 million and above in urban areas, medium-sized cities with a resident population of 500,000–1 million, and small-sized cities with a resident population of 500,000 and below in urban areas

*** indicates $P < 0.01$

** indicates $P < 0.05$, and

* indicates $P < 0.1$. 2000 ∪ 2015 is the set of urban labor migration rate and urban skill relatedness in 2000 and 2015.

insignificant in medium- and small-sized cities, their positive values above zero still reflect a positive effect between the two.

Overall, the results show that the skill relatedness of an occupational class to a city significantly facilitates the labor migration of that occupational class in that city. This facilitating effect increases with the degree of skill relatedness of the occupational class to the city. When the skill-relatedness of the occupational class to the city enters an extremely strong state, its facilitating effect on labor migration is more significant. This significance is most pronounced in cities with high-size classes, followed by small- and medium-sized cities. Accordingly, the hypothesis of the effect of skill association on the location choice of heterogeneous labor force is tested empirically. The heterogeneous labor force chooses to move into areas with strong skill relatedness or relatedness owing to the class division effect and moves out or does not move into areas with no skill relatedness owing to the selection effect. The empirical results validate the hypotheses presented in the previous section.

## Discussion and conclusions

This paper divided the occupations of the labor force into 13 occupational subcategories and clustered into three skill levels based on labor skill heterogeneity. We analyzed the spatiotemporal characteristics of heterogeneous labor location choices based on cross-sectional data from 288 prefecture-level cities in China in 2000 and 2015. Moreover, this study explored the relationship between skill associations and location choices of the labor force divided into occupational subcategories. Hence, we discussed how spatial class division and selection effects endogenously affected heterogeneous labor location decisions and examined the variability of the impact of high-skill association on heterogeneous labor migration across cities of different size classes. This study has important implications for the research on skill relatedness and heterogeneous labor location choice. Our empirical results have three main findings.

First, our findings indicate that labor mobility in China increased significantly from 2000 to 2015, with significant and widening regional differences in labor migration rates across occupational categories between cities and increased regional concentration of labor location choice. The intensity of inter-city population mobility is an essential indicator of regional economic relations, urban hierarchy, and network structure strength [61]. The widening of regional differences in heterogeneous labor force mobility from 2000 to 2015 also reflects China's increased imbalance of regional development [62]. Meanwhile, our study identified six significant labor agglomeration location choices for each occupational category: Beijing-based Bohai Rim, Shanghai-based Yangtze River Delta, West Coast, Pearl River Delta, Hainan, and Xinjiang and Tibet-based Western agglomerations. Note that these regions are developing rapidly relative to their surrounding areas. The dominance of the central cities in the region is extreme (e.g., Beijing and Shanghai), and the result of this phenomenon may cause the urban scale of major cities to expand, triggering a certain degree of "urban disease". Surrounding cities are constantly deprived of development resources and opportunities, and their development is restricted [63]. The solution to this problem can be achieved by the diffusion of core city functions [64]. In addition, changes in the number of high-migration and higher-migration cities for different occupational groups of the labor force relatively indicate the changes in the development stages of other cities and corresponding changes in the demand for different occupational groups of the labor force.

Second, our findings emphasize that skill relatedness positively affects heterogeneous labor location choice in Chinese prefecture-level cities. The effect on labor location choice is more significant when the skill-relatedness of a city is more vigorous. That is, the stronger the skill relatedness to regional employment, the higher the skill proximity, and the closer the skill

distance, the more employment opportunities for the related labor force in that city and the greater the likelihood of its migration. Our analysis highlights the impact of skill relatedness on the location choice of a heterogeneous labor force. It provides a new perspective for the study of location choice of the heterogeneous labor force. Furthermore, our findings suggest that the magnitude of the effect of high-skill relatedness on labor location choice varies across cities of different size classes. The impact of high-skill relatedness on labor location choice is also higher in cities with higher size classes. The findings echo our analysis of the spatiotemporal characteristics of heterogeneous labor location choice. That is, cities with high-size courses in the region are more likely to attract labor to them.

Lastly, the labor force with a skill-relatedness or strong relatedness to regional employment moves to the location owing to the class division effect. The labor force with no skill-relatedness or weak relatedness moves out or does not move to the area due to the selection effect. Our analysis generally places the skill attribute characteristics of micro-individuals of the heterogeneous labor force in the same research framework with the skill structure characteristics of the macro-region, considers skill relatedness as the mechanism of spatial class differentiation and selection, and interprets the location selection of heterogeneous labor force with skill relatedness as the core. It fills in the gap in the empirical analysis of how spatial categorization and selection endogenously affect heterogeneous labor location decisions in studies of spatial economics. Meanwhile, our findings have implied countermeasure suggestions. Regions should vigorously attract labor with high skill relatedness to the local occupational space to improve the labor market match and stabilize regional employment growth. Note that blind competition for talent is undesirable. In recent years, cities have introduced preferential policies for high-skilled labor. Nevertheless, our study indicates that the policy effect will be unsustainable if the attracted high-skilled labor is not strongly associated with the skills of employees in the city, coupled with the subsequent gradual weakening of administrative instruments. The weakly associated labor will leave the city owing to the selection effect. This result aligns with previous studies, which have shown that cities' talent policies in the "war of attracting talent" lack a compelling impact [65].

Our study has the following limitations. First, this study can only choose 2000 and 2015 for the time points of the research owing to data availability, pending further research based on the 7th China Census microdata set to ensure the timeliness of our study. Second, many complex factors limit and influence population migration. This study only controlled for some influencing factors from the two perspectives of class effects and selection effects. We obtained some interesting results, but there may be other influencing factors between the independent and dependent variables of the model. Therefore, we should continue to think about the choice of control variables in future studies. Third, this paper studies the relationship between skill affinity and labor mobility. There may be an endogenous relationship between these two variables. We considered adding instrumental variables to determine causality but did not find suitable ones. Therefore, we will consider eliminating the inherent quality problem in the follow-up research. In addition, our study uses the skilled labor migration rate as the dependent variable. Given that the micro data set provides sampled data, some municipalities' skilled labor migration is 0. This situation also makes the dependent variable a typical truncated data. Therefore, further research should estimate the effect of skill association on heterogeneous labor location choice using other models to improve the accuracy of the results.

## Author Contributions

**Conceptualization:** Xiaoqi Zhou, Rongjun Ao, Yierfanjiang Aihemaitijiang.

**Formal analysis:** Yierfanjiang Aihemaitijiang.

**Funding acquisition:** Rongjun Ao.

**Methodology:** Xiaoqi Zhou, Rongjun Ao, Yierfanjiang Aihemaitijiang.

**Supervision:** Hui Tang.

**Validation:** Hui Tang.

**Visualization:** Yierfanjiang Aihemaitijiang.

**Writing – original draft:** Xiaoqi Zhou.

**Writing – review & editing:** Xiaoqi Zhou, Jing Chen.

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
