## [Decision Letter · Decision Letter 0]

2 May 2023

PONE-D-23-09714Influence of skill relatedness on the location choice of heterogeneous labor force in Chinese prefecture-level citiesPLOS ONE

Dear Dr. aihemaitijiang,

Thank you for submitting your manuscript to PLOS ONE. After careful consideration, we feel that it has merit but does not fully meet PLOS ONE’s publication criteria as it currently stands. Therefore, we invite you to submit a revised version of the manuscript that addresses the points raised during the review process. Authors need to address comments of reviewers. Add research objectives and significant importance of this research. What is contribution to existing knowledge.  Please submit your revised manuscript by Jun 16 2023 11:59PM. If you will need more time than this to complete your revisions, please reply to this message or contact the journal office at plosone@plos.org. Please include the following items when submitting your revised manuscript:A rebuttal letter that responds to each point raised by the academic editor and reviewer(s). You should upload this letter as a separate file labeled 'Response to Reviewers'.A marked-up copy of your manuscript that highlights changes made to the original version. You should upload this as a separate file labeled 'Revised Manuscript with Track Changes'.An unmarked version of your revised paper without tracked changes. You should upload this as a separate file labeled 'Manuscript'.

We look forward to receiving your revised manuscript.

Kind regards,

Muhammad Tayyab Sohail

Academic Editor

PLOS ONE

Journal Requirements:

"This research was funded by the National Natural Science Foundation of China (Grant numbers 42271188)."

4. We note that Figures 2 and 3 in your submission contain map/satellite images which may be copyrighted. All PLOS content is published under the Creative Commons Attribution License (CC BY 4.0), which means that the manuscript, images, and Supporting Information files will be freely available online, and any third party is permitted to access, download, copy, distribute, and use these materials in any way, even commercially, with proper attribution. For these reasons, we cannot publish previously copyrighted maps or satellite images created using proprietary data, such as Google software (Google Maps, Street View, and Earth). For more information, see our copyright guidelines: http://journals.plos.org/plosone/s/licenses-and-copyright.

a. You may seek permission from the original copyright holder of Figures 2 and 3 to publish the content specifically under the CC BY 4.0 license.  

Reviewers' comments:

Reviewer's Responses to Questions

**Comments to the Author**

1. Is the manuscript technically sound, and do the data support the conclusions?

Reviewer #1: Yes

Reviewer #2: Yes

Reviewer #3: Yes

2. Has the statistical analysis been performed appropriately and rigorously? 

Reviewer #1: Yes

Reviewer #2: Yes

Reviewer #3: Yes

3. Have the authors made all data underlying the findings in their manuscript fully available?

Reviewer #1: Yes

Reviewer #2: Yes

Reviewer #3: Yes

4. Is the manuscript presented in an intelligible fashion and written in standard English?

Reviewer #1: Yes

Reviewer #2: Yes

Reviewer #3: Yes

5. Review Comments to the Author

Reviewer #1: Dear Editors:

I have reviewed the paper titled “Influence of skill relatedness on the location choice of heterogeneous labor force in Chinese prefecture-level cities”. Based on skill-related assumptions, the authors interpret the cross-regional migration of heterogeneous labor in China. The perspective given in the paper contributes to economic geography, especially by combining cross-regional migration of population and geographic space to some extent. However, I think some major revisions are still needed before accepting this article.

1. Table 1 reports the clustering methodology and specific classifications for occupational categories. When presenting the methodology (in the Data description and data sources sections), the authors seem to focus only on presenting the occupational codes used in the census/sample survey but do not specifically present the reasons for analyzing skill correlations in this way. Thus, the rationale for the distinction of agriculture-related jobs as low-skilled needs to be further described.

2. The paper examines the relationship between skill relatedness and labor migration rates, but there appears to be an endogenous relationship between the two variables that cannot be ignored. Even though the authors may agree that the endogenous relationship between the two is mutual, a more precise analytical framework should be adopted for consideration. I suggest that the authors use methods that eliminate endogeneity such as instrumental variables to identify causality.

3. The paper uses the Gini coefficient several times (e.g., line 293). As far as I know, no official Gini coefficient indicator has been published for prefecture-level cities and towns in China, while the results may vary from measurement to measurement. I think the authors should explain this indicator's calculation method or data source.

4. The analytical framework presented in Figure 1 is currently seen as independent, and I would like the authors to present it more fully. And focus on answering the question of why people with high-skilled relatedness are more likely to emigrate, while people with low-skilled relatedness immigrate. This requires clarifying a research perspective, which is whether the authors are concerned with the direction of labor mobility. Because the current framework given is different from the usual thinking of us, it would be more interesting to explain this point clearly.

5. The results of the two types of heterogeneity tests reported in Table 5 differ in terms of the sample size of the data. The total sample for the skill relativeness subgroup is 7488, while the total sample for the city size subgroup is 2563. why?

6. I recommend that authors read and refer to the following literature:

(1) Gu, H., Meng, X., Shen, T., & Wen, L. (2020). China’s highly educated talents in 2015: Patterns, determinants and spatial spillover effects. Applied Spatial Analysis and Policy, 13(3), 631-648.

(2) Lao, X., Gu, H., Yu, H., & Xiao, F. (2021). Exploring the spatially-varying effects of human capital on urban innovation in China. Applied Spatial Analysis and Policy, 14(4), 827-848.

Minor problems:

1. I think the position of “low-skilled” and “high-skilled” in Table 1 seems to be reversed, which is contrary to the article, please double-check.

2. There are two “individual fixed effects” in Tables 4 and 5, is there any difference between them? If it is a two-factor fixed effects regression model, I suggest that the authors specify the names of the two types of fixed effects.

3. I advise the author to seek help from native English speakers, existing expressions need to be improved.

Reviewer #2: Title: Influence of skill relatedness on the location choice of heterogeneous labor force in Chinese prefecture-level cities

This paper use labor force migration data and employee data by occupation subcategory in prefecture-level cities from the China 2000 NationalPopulation Census and China 2015 National Population Sample Survey microdata

sets.Some suggestions for improvement are as follows:

What is the author's theoretical background and foundation. The authors affirm the use of the classical migration model, so what are the authors' extensions and applications of this model in those areas.

The authors simply need to answer why the use of data up to 2015, the latest data seems to be available. There is no need for the author to update the data, just to explain.

The economic motivation behind the results needs to be explained.

Reviewer #3: Many thanks to the editor for the invitation. In general, the author's research is very new and by the times relevant. I would like the authors to address the following questions. In line 84, the author says that despite some current limitations ，，，，. A moderate review of the literature and discussion is needed here, otherwise she is lacking basis.Spatial econometric models need to satisfy a variety of conditions. How the authors validate them.

6. PLOS authors have the option to publish the peer review history of their article (what does this mean?). If published, this will include your full peer review and any attached files.

Reviewer #1: No

Reviewer #2: No

Reviewer #3: No

---

## [Author Response · Author response to Decision Letter 0]

8 Jun 2023

Dear Reviewer,

Thank you for your letter. Those comments are all valuable and very helpful for revising and improving our paper and the important guiding significance to our research. We have studied comments carefully and have made a correction which we hope meet with approval. We used "Track Changes" in the paper. The main corrections in the paper and the responses to your comments are as following:

Reviewer #1: 

I have reviewed the paper titled “Influence of skill relatedness on the location choice of heterogeneous labor force in Chinese prefecture-level cities”. Based on skill-related assumptions, the authors interpret the cross-regional migration of heterogeneous labor in China. The perspective given in the paper contributes to economic geography, especially by combining cross-regional migration of population and geographic space to some extent. However, I think some major revisions are still needed before accepting this article.

1.Table 1 reports the clustering methodology and specific classifications for occupational categories. When presenting the methodology (in the Data description and data sources sections), the authors seem to focus only on presenting the occupational codes used in the census/sample survey but do not specifically present the reasons for analyzing skill correlations in this way. Thus, the rationale for the distinction of agriculture-related jobs as low-skilled needs to be further described.

Response:

Thanks for pointing it out. We have added the reasons for measuring skills by occupation in the first paragraph of this section and added relevant references. The supplemental sections are as follows:

Skills are not uniformly defined. Although educational attainment has been used to measure skill level, a growing number of researchers believe that educational attainment is not the best indicator of skills and propose to measure skills by occupation. Ingram et al.[48] point out that years of education are a rough measure of skills because a degree is not equivalent to the talent it implies, and all students, even those who graduate from the same school with the same degree, may not graduate at the same level. Florida et al.[49] argue that formal education can only measure potential talents or skills, while occupations provide an idea of "how human talents or abilities are absorbed and utilized by the economy." Based on existing research, this paper uses occupation as the primary indicator to measure skills, considering that skills reflect the ability to complete a job.

In addition, the reason for classifying skill levels is mainly based on the overall nature of work and job skill requirements of each occupational group, after mapping Chinese occupational groups to international occupational classifications, as described in the text.

2.The paper examines the relationship between skill relatedness and labor migration rates, but there appears to be an endogenous relationship between the two variables that cannot be ignored. Even though the authors may agree that the endogenous relationship between the two is mutual, a more precise analytical framework should be adopted for consideration. I suggest that the authors use methods that eliminate endogeneity such as instrumental variables to identify causality.

Response:

Thank you for your question. What you say is quite reasonable. Therefore, we considered the method of selecting instrumental variables to eliminate endogeneity but encountered some difficulties. The main reason is that we are cross-sectional data, so we cannot lag the first-phase variable as the instrumental variable. Besides, the sample number of main explanatory variables is large, so we can't find suitable instrumental variables for regression analysis. In this case, we put this problem into the discussion section, hoping to solve it in future research. The relevant discussion sections are as follows:

Our study has the following limitations. First, this study can only choose 2000 and 2015 for the time points of the research owing to data availability, pending further research based on the 7th China Census microdata set to ensure the timeliness of our study. Second, many complex factors limit and influence population migration. This study only controlled for some influencing factors from the two perspectives of class effects and selection effects. We obtained some interesting results, but there may be other influencing factors between the independent and dependent variables of the model. Therefore, we should continue to think about the choice of control variables in future studies. Third, this paper studies the relationship between skill affinity and labor mobility. There may be an endogenous relationship between these two variables. We considered adding instrumental variables to determine causality but did not find suitable ones. Therefore, we will consider eliminating the inherent quality problem in the follow-up research. In addition, our study uses the skilled labor migration rate as the dependent variable. Given that the micro data set provides sampled data, some municipalities’ skilled labor migration is 0. This situation also makes the dependent variable a typical truncated data. Therefore, further research should estimate the effect of skill association on heterogeneous labor location choice using other models to improve the accuracy of the results.

3.The paper uses the Gini coefficient several times (e.g., line 293). As far as I know, no official Gini coefficient indicator has been published for prefecture-level cities and towns in China, while the results may vary from measurement to measurement. I think the authors should explain this indicator's calculation method or data source.

Response:

Thanks for the suggestion. At present, it is true that China has not published the official Gini coefficient index. In this paper, the Gini coefficient is mainly used with the coefficient of variation to discuss the regional concentration of labor migration rate. The relevant expression and reference calculation method can be found in line 254-256.

4.The analytical framework presented in Figure 1 is currently seen as independent, and I would like the authors to present it more fully. And focus on answering the question of why people with high-skilled relatedness are more likely to emigrate, while people with low-skilled relatedness immigrate. This requires clarifying a research perspective, which is whether the authors are concerned with the direction of labor mobility. Because the current framework given is different from the usual thinking of us, it would be more interesting to explain this point clearly.

Response:

Thank you for your question. We explain the framework in Figure 1 in more detail. Although we have focused on the direction of labor mobility, the focus is still on incorporating skill relatedness into the process of spatial classification and selection of heterogeneous subjects. The relevant sections are as follows:

The analytical framework centred on spatial sorting, and selection effect can be used to analyze the location choice of a heterogeneous labor force. The drivers of spatial sorting and selection of heterogeneous subjects include various economic and non-economic factors [26-32]. The push-pull theory framework also consists of the regional push and pull factors. Only the push-pull factors under the push-pull theory are consistent for all labor forces. In contrast, spatial class sorting and choice theory can consider the variability of push-pull factors for different labor force individuals. The impact of skill relatedness on skill heterogeneous labor location choice can also be interpreted with the help of this analytical framework. In particular, the spatial sorting effect refers to the choice of cities by micro-subjects, an ex-ante locational choice made by heterogeneous subjects based on their talents [36,37]. The spatial selection effect is the city's choice of micro issues, manifested by eliminating subjects with lower ability owing to market competition or cost of living, leaving the most talented ones behind afterwards [34,38]. The effect of skill relatedness on the locational choice of the heterogeneous labor force in Chinese prefecture-level cities is shown in Fig 1. Suppose the labor force is related (similar or complementary) to a city's existing employment skill structure. In the case of increased relevant employment opportunities in that city, the labor force may migrate to that city based on the matching skills they possess, thus meeting the city's labor demand. In cases such as industrial structure changes, when the skill attributes of the city's existing workforce do not match the regional labor market demand, there will be a workforce that is priced out of the market and chooses to leave the city because it cannot find a job. The effect of spatial sorting and selection is a spontaneous process of spatial allocation caused by the pairing between the labor force and cities. Accordingly, Hypothesis 3 is proposed.

5.The results of the two types of heterogeneity tests reported in Table 5 differ in terms of the sample size of the data. The total sample for the skill relativeness subgroup is 7488, while the total sample for the city size subgroup is 2563. why?

Response:

We thank the reviewer for this insightful comment. We rechecked the calculations. Our skill relatedness subgroup was 7488, the city size subgroup was a further analysis of the strong skill relatedness sample with a total sample size of 2500, and the wrong part in the table was the sample size of small cities, which should be 437.

6.I recommend that authors read and refer to the following literature:

(1) Gu, H., Meng, X., Shen, T., & Wen, L. (2020). China’s highly educated talents in 2015: Patterns, determinants and spatial spillover effects. Applied Spatial Analysis and Policy, 13(3), 631-648.

(2) Lao, X., Gu, H., Yu, H., & Xiao, F. (2021). Exploring the spatially-varying effects of human capital on urban innovation in China. Applied Spatial Analysis and Policy, 14(4), 827-848.

Response:

Thank you for the literature recommendation. These two articles, An article examines the distribution, driving forces, and spatial effect of highly educated talents at the prefecture level. Another concern is the spatially-varying effects of human capital on urban innovation. This paper suggests that different policies should be formulated in cities of diverse conditions that are sensitive to their contexts. We have benefited a lot from reading the article and have quoted the two papers in our manuscript to increase the credibility and persuasiveness of the manuscript.

7.Minor problems:

(1)I think the position of “low-skilled” and “high-skilled” in Table 1 seems to be reversed, which is contrary to the article, please double-check.

(2)There are two “individual fixed effects” in Tables 4 and 5, is there any difference between them? If it is a two-factor fixed effects regression model, I suggest that the authors specify the names of the two types of fixed effects.

(3)I advise the author to seek help from native English speakers, existing expressions need to be improved.

Response:

Thanks for pointing it out. First, upon review, the position of "low-skilled" and "high-skilled" in table 1 have indeed been reversed and have been revised. Second, the two "individual fixed effects" in original tables 4 and 5 were erroneous and should have been "individual fixed effects" and "time fixed effects", respectively, which have been corrected. In addition, we sought native English speakers to polish our manuscript to better meet the reader's reading requirements.

Reviewer #2: 

This paper use labor force migration data and employee data by occupation subcategory in prefecture-level cities from the China 2000 National Population Census and China 2015 National Population Sample Survey microdata sets. Some suggestions for improvement are as follows:

1.What is the author's theoretical background and foundation. The authors affirm the use of the classical migration model, so what are the authors' extensions and applications of this model in those areas.

Response:

Our theoretical basis and background are mainly the relatedness law of economic spatial evolution, the spatial classification and selection theory of heterogeneous subjects and the push and pull theory of population migration. Therefore, our literature review revolves around relatedness, especially skill relatedness, and spatial choices for labor. 

We affirm the use of classical migration models, which geographers use to influence the location choice of labour by various economic and non-economic factors. However, these factors are more about the effect on the labour force as a whole. At the same time, the micro-mechanism of spatial classification and selection theory can consider the differences of different influencing factors on different labour force individuals. In addition, the study of relatedness rule points out that the labour force with skills associated with regional employment has more employment opportunities in the region, and skill relatedness is also an important driving factor for the spatial classification and selection of heterogeneous labour force. The stronger the skill relatedness with the overall regional employment, the higher the skill proximity, the closer the skill distance, and the greater the possibility of labour force migration. The existing classical migration model does not include the micro individual labour force and the macro overall regional employment in the research framework. The theoretical framework of skill relatedness can solve this problem. Therefore, with the help of skill relatedness, our research extension attempts to substantially explain how spatial class classification and choice endogenously influence the location of heterogeneous labour.

We explain this in the last two paragraphs of the introduction and add further in the last paragraph of the Research hypotheses.

2.The authors simply need to answer why the use of data up to 2015, the latest data seems to be available. There is no need for the author to update the data, just to explain.

Response:

As far as we know, labor migration data for occupational subclasses in prefecture-level cities in China can only be obtained from microdata from the 2000 census and the 2015 population sample survey. The data from the sixth census in 2010 would have been a very good data source, but the labor migration data in this dataset is not detailed to the level of prefecture-level urban occupational subcategories. Our team is applying for and exploring the 7th Population Census microdata set in 2020, which is not yet available. We hope to update the data in subsequent studies to ensure that the studies are time-sensitive.

3.The economic motivation behind the results needs to be explained.

Response:

Thanks for the suggestion. Our explanation of the economic motivation behind the results is mainly reflected in the Discussion and Conclusions section. We mainly obtained three research results and discussed them separately.

First, the widening regional differences in the heterogeneity of labor mobility reflect the intensification of regional development imbalance in China, and the study obtains six important labor agglomeration locations, which provokes us to think about the relationship between central cities and peripheral cities. At the same time, the changes in the number of high-migration and higher-migration cities of different occupational labor also indicate to a certain extent that different urban development stages have changed, and the demand for different occupational labor has also changed accordingly.

Second, our findings highlight the positive impact of skills correlation on heterogeneous labor siting in prefecture-level cities in China. When skill relatedness is more active in a city, the impact on labor site selection is more pronounced. That is, the stronger the correlation between skills and regional employment, the higher the proximity of skills, and the closer the distance of skills, the more employment opportunities for the relevant labor force in the city, and the greater the likelihood of its migration.

Third, we propose policy recommendations based on the findings of categorization and selection effects. That is, each region should vigorously attract high-skilled labor related to local occupational space, improve the matching degree of labor market, and stabilize regional employment growth. And pointed out that blind competition for talent is not desirable.

Reviewer #3: 

Many thanks to the editor for the invitation. In general, the author's research is very new and by the times relevant. I would like the authors to address the following questions. 

1.In line 84, the author says that despite some current limitations,,,. A moderate review of the literature and discussion is needed here, otherwise she is lacking basis.

Response:

Thanks for the suggestion. We have supplemented the previously mentioned literature in the relevant section so that the limitation of the summary of this paragraph is more scientific.

2.Spatial econometric models need to satisfy a variety of conditions. How the authors validate them.

Response:

Thanks for pointing it out. We have made corresponding supplements in the front of the model formula. Given the length of the manuscript, we add the following:

For model selection, the ordinary least squares method (OLS) is used to evaluate the relationship between two or more factor attributes and minimize the sum of squares of residuals of all observed values in the selected global regression model. In this study, variables were first standardized to eliminate the multicollinearity problem, and then the OLS model was used to analyze the cross-section data of 2000 and 2015 preliminarily. Then, a two-factor fixed effects model was used for the panel data consisting of 288 cities in both years. This study used the F, and Hausman tests to determine specific model selection when establishing the panel data model. Since the P-value of the F test was 0, the Hausman test accepted the null hypothesis that explanatory variables were related to individual (or time) effects, so the fixed effects model was chosen. Moreover, the two-factor fixed effects model was selected because of the highest confidence and goodness of fit of the estimated results of the two-factor fixed effects model. The regression equation is as follows.

Thank you once again for the opportunity to comment on this manuscript. The paper has improved much and is moving towards a stronger contribution. Best wishes.

---

## [Decision Letter · Decision Letter 1]

27 Jul 2023

Influence of skill relatedness on the location choice of heterogeneous labor force in Chinese prefecture-level cities

PONE-D-23-09714R1

Dear Dr. aihemaitijiang,

We’re pleased to inform you that your manuscript has been judged scientifically suitable for publication and will be formally accepted for publication once it meets all outstanding technical requirements.

Kind regards,

Muhammad Tayyab Sohail

Academic Editor

PLOS ONE

Additional Editor Comments (optional):

Reviewers' comments:

Reviewer's Responses to Questions

**Comments to the Author**

1. If the authors have adequately addressed your comments raised in a previous round of review and you feel that this manuscript is now acceptable for publication, you may indicate that here to bypass the “Comments to the Author” section, enter your conflict of interest statement in the “Confidential to Editor” section, and submit your "Accept" recommendation.

Reviewer #2: All comments have been addressed

2. Is the manuscript technically sound, and do the data support the conclusions?

Reviewer #2: Yes

3. Has the statistical analysis been performed appropriately and rigorously? 

Reviewer #2: Yes

4. Have the authors made all data underlying the findings in their manuscript fully available?

Reviewer #2: Yes

5. Is the manuscript presented in an intelligible fashion and written in standard English?

Reviewer #2: Yes

6. Review Comments to the Author

Reviewer #2: Thanks to the reviewer for the invitation. The authors have responded to the review comments have been point by point and given full points to validate it. Overall, the paper has been adequately revised and I recommend its acceptance.

7. PLOS authors have the option to publish the peer review history of their article (what does this mean?). If published, this will include your full peer review and any attached files.

Reviewer #2: No

---

## [Editor Report · Acceptance letter]

16 Aug 2023

PONE-D-23-09714R1 

Influence of skill relatedness on the location choice of heterogeneous labor force in Chinese prefecture-level cities 

Dear Dr. Aihemaitijiang:

I'm pleased to inform you that your manuscript has been deemed suitable for publication in PLOS ONE. Congratulations! Your manuscript is now with our production department. 

Kind regards, 

on behalf of

Dr. Muhammad Tayyab Sohail 

Academic Editor

PLOS ONE